# Effectiveness of Restoration Treatments for Reducing Fuels and Increasing Understory Diversity in Shrubby Mixed-Conifer Forests of the Southern Rocky Mountains, USA

**Julie E. Korb** [1], **Michael T. Stoddard** [2,*] and **David W. Huffman** [2]

1    Department of Biology, Fort Lewis College, 1000 Rim Drive, Durango, CO 81301, USA;
     KORB_J@fortlewis.edu
2    Ecological Restoration Institute, Northern Arizona University, P.O. Box 15017, Flagstaff, AZ 86011, USA;
     david.huffman@nau.edu
*    Correspondence: mike.stoddard@nau.edu

**Abstract:** Exclusion of natural surface fires in warm/dry mixed-conifer forests of the western U.S. has increased potential for stand-replacing crown fires and reduced resilience of these systems to other disturbances, such as drought and insect attack. Tree thinning and the application of prescribed fire are commonly used to restore more resilient ecological conditions, but currently, there is a lack of long-term data with which to evaluate restoration treatment effectiveness in forest types where resprouting shrubs dominate understory communities. At a mixed-conifer site in southwestern Colorado, we compared forest structure and understory vegetation responses to three restoration treatments (thin/burn, burn, and control) over 10 years in a completely randomized and replicated experiment. Forest density, canopy cover, and crown fuel loads were consistently lower, and crown base height was higher, in thin/burn than burn or controls, but the effects diminished over time. Ten years following treatment, >99% of all plant species within both treatments and the control were native in origin. There were no differences between treatments in understory richness, diversity, cover, or surface fuels, but graminoid cover more than doubled in all treatments over the 15-year monitoring period. Similarly, there was more than a 250% increase post-treatment in shrub density, with the greatest increases in the thin/burn treatment. In addition, we saw an increase in the average shrub height for both treatments and the control, with shrub stems >80 cm becoming the dominant size class in the thin/burn treatment. Conifer seedling density was significantly lower in thin/burn compared with burn and control treatments after 10 years. Taken together, these conditions create challenges for managers aiming to reestablish natural fire patterns and sustain mixed-conifer forests. To limit the dominance of resprouting shrubs and facilitate conifer regeneration after overstory thinning and prescribed fire, managers may need to consider new or more intensive approaches to forest restoration, particularly given current and projected climate change.

**Keywords:** crown fire hazard; mechanical thinning; prescribed fire; herbaceous understory; resprouting shrubs; tree regeneration

## 1. Introduction

Mixed-conifer forests in the southwest occur along a continuum from warm/dry to cool/mesic [1]. Warm/dry mixed-conifer forests are dominated by fire-resistant species, such as ponderosa pine (*Pinus ponderosa* Lawson & C. Lawson) and Douglas fir (*Pseudotsuga menziesii* (Mirb.) Franco)), but also include mesic species, such as white fir (*Abies concolor* (Gord. & Glend.) Lindl. ex Hildebr.) and

aspen (*Populus tremuloides* Michx.). This forest type is located at elevations (2250–3000 m) just above ponderosa pine forests. Over the last few decades, there has been an increase in wildfire size and behavior in dry mixed-conifer forest types across the western United States, with a shift towards uncharacteristically large (>1000 ha) stand-replacing crown fires [2,3]. Frequent surface fires are an important aspect of natural disturbance regimes in dry mixed-conifer systems, and stand-replacing crown fires are not the norm [4]. In southwestern Colorado, understory and midstory communities are variable but can be dominated by persistent resprouting shrubs and hardwood trees, such as Gambel oak (*Quercus gambelii* Nutt.), serviceberry (*Amelanchier alnifolia* (Nutt.) Nutt. ex M. Roemer), and snowberry (*Symphoricarpus rotundifolius* Gray).

Prior to Euro-American settlement, dry mixed-conifer forests primarily burned as low to moderate severity surface fires but also experienced mixed severity fires that included surface, torching, and crowning depending on the topography, fuels, and climate, with mean fire return intervals (MFIs) of 2–32 years [5–7]. Dry mixed-conifer forests were historically heterogeneous in structure due to diverse fire behavior, which promoted ecological resilience and understory species diversity [8]. Due to the past management activities of fire suppression, logging, and grazing, dry mixed-conifer forests are increasingly homogeneous with higher fuel loads, and the composition shifted toward shade-tolerant and fire-intolerant species [9–11]. Climatic warming, drought, and forest structural changes have resulted in decreased ecosystem resilience and increased susceptibility to stand-replacing crown fires in these as well as other dry mixed-conifer forest types across the west [4,12–14]. In response, land managers over the past two decades have used mechanical tree thinning and prescribed fire to decrease the susceptibility of crown fire and restore more characteristic forest structure, composition, and ecological function.

Despite the widespread use of forest restoration treatments in dry forest types across the western U.S. for the past few decades, questions remain regarding treatment effects on ecosystem dynamics due to the scarcity of long-term (≥10 years) multi-scaled studies [15]. For example, increasing native plant diversity is one of the most cited goals for ecological restoration [16], but presently, there is a dearth of research on how treatments affect understory plant communities over the long term [17–22]. In a meta-analysis, Schwilk et al. [23] found no consistent pattern in understory community diversity and abundance in response to forest restoration treatments. However, Abella and Springer [20] found that understory abundance consistently increased in studies that monitored changes for five years or longer. On sites where sprouting woody species are abundant in the understory or midstory, rapid increases following tree thinning and fire may inhibit the establishment and growth of other less competitive species [11]. These patterns may in turn affect ecosystem dynamics, particularly with subsequent severe fire [24]. Further, the effects of shrub responses on the treatment longevity and when follow-up treatments are necessary remains unclear for land managers [11,25–27]. Pretreatment stand conditions, treatment intensity, and site productivity all affect the longevity of restoration treatment effectiveness [28].

In 2002, we initiated a field experiment to compare no treatment with two forest restoration treatments (thinning followed by prescribed fire, prescribed fire only, and no treatment) commonly applied to reduce hazardous fuels, increase understory diversity, and increase resilience to climate and disturbance. We monitored ecological responses, 1, 5, and 10 years following treatments in dry mixed-conifer forests of southern Colorado and asked the following questions:

(1) How do forest structure and hazardous fuels vary among the restoration treatments?

(2) How do understory plant richness, diversity, abundance, and community dynamics differ among treatments over time? and

(3) Do conditions 10 years post-treatment indicate enhanced resiliency?

## 2. Materials and Methods

### 2.1. Study Area

Our study area is located on lower Middle Mountain in the San Juan Mountains (N 37.296, W 107.228) on the San Juan National Forest approximately 18 km northwest of Pagosa Springs, in southwest Colorado. Lower Middle Mountain consists of moderately steep (15%–30%) slopes on generally south-facing aspects. Elevations range from 2438 to 2743 m. The dominant soil type is Dutton loam, a silty clay loam [29]. Average daily temperatures range from a maximum of 28.2 °C in July to a minimum of −17 °C in January. Average annual precipitation is 55.0 cm, with the greatest amounts occurring in July and August. Snow dominates precipitation from November to March, with an average annual total snowfall of 295.7 cm [30].

A large portion of the warm/dry mixed-conifer forest on Lower Middle Mountain has numerous old-growth characteristics, which include large trees, spike-topped or broken-topped trees, snags, and large coarse woody debris. Forest vegetation is dominated by ponderosa pine, Douglas fir, and white fir interspersed with small pockets of both mature and young aspen (*P. tremuloides* Michx.) The midstory and understory are dominated primarily by white fir and Douglas fir, with a variety of shrubs, including Gambel oak (*Q. gambelii* Nutt.), snowberry, and serviceberry. Ponderosa pine regeneration is present. Common herbaceous species at the site include blue wild rye (*Elymus glaucus* Buckley), Thurber's fescue (*Festuca thurberi* Vasey), Parry's oatgrass (*Danthonia parryi* Scribn.), muttongrass (*Poa fendleriana* (Steud.) Vasey), little sunflower (*Helianthella quinquenervis* (Hooker) Gray), tuber starwort (*Pseudostellaria jamesiana* (Torrey) Weber and Hartman), and showy fleabane (*Erigeron speciosus* (Lindley) de Candolle).

Past disturbance history on Lower Middle Mountain includes sheep grazing beginning in the late 1800s, cattle grazing since the early 1900s, and fire suppression since the early 20th century. There are no large meadows or water hole-tanks and minimal grazing within the area. A single timber harvest occurred between 1990 and 1993, which was the first harvest of an intended three-harvest shelterwood cut to manage the stand for even-aged conditions. Foresters removed 3830 million board feet, which consisted of 51% ponderosa pine, 33% white fir, and 16% Douglas fir evenly across the study area [29].

### 2.2. Experimental Design and Field Methods

Thinning prescription was based on site-specific reconstructed historical forest structure and retained all living trees that established in 1870 or earlier as identified by size, bark color, and canopy architecture [6]. We also retained live post-settlement trees as substitutes for remnants (e.g., snags, logs, and stumps) if the remnants were present in 1870 but were no longer alive. On average, we kept two younger trees of the same species, within 20 m of each dead remnant (see Korb et al. [31] for a detailed description of the thinning prescription). Logs or limbs were not removed following thinning treatments in 2004, nor did we rake around old-growth trees to remove fuels around tree boles. Fire crews did prescribe burning in fall 2007 (Blocks 1 and 2) and fall 2008 (Blocks 3 and 4) using strip head fires. Fire crews were unable to burn all units during the same time period due to smoke output regulations. Average flame lengths were 0.3–0.9 m in needle duff, 1–2.4 m in the tree thinning slash, and up to 7.6 m from torching trees, with a very low rate of spread for both years. Post-treatment forest measurements were conducted in 2018 as described in Korb et al. [31] and Stoddard et al. [26]. We established four replicate blocks of three randomly assigned treatment units (~16 ha/unit): 1) Thinning followed by prescribed fire (thin/burn), 2) prescribed fire only (burn), and 3) no treatment (control). We established 20 permanent monitoring plots to characterize forest structure and vegetation on a 60-m grid based on a systematic starting grid point (total N = 4 blocks × 3 treatment units/block × 20 plots/unit = 240 plots). Plot centers were permanently marked with iron stakes and georeferenced with global positioning systems. We collected pre-treatment data in the summer of 2003 and post-treatment data in the summers of 2009, 2013, and 2018. Overstory trees and saplings taller than breast height (137 cm) were measured in a 400-m$^2$ (11.28 m radius) circular plot.

For each tree in every plot, we recorded species, condition (living or snag/log classes Thomas et al. [32]), diameter at breast height (dbh), total height, and crown base height. We tallied tree regeneration for seedlings (<40 cm in height and ≤2.5 cm DBH), saplings (>40.1 cm in height and ≤2.5 cm DBH), and individual shrub stems by species, condition, and height class (<40 cm; 40.1–80 cm; 80.1–137 cm) on a nested 100-m² (5.64 m radius) subplot. Tree canopy cover was recorded using a vertical projection densitometer every 3 m along a permanently marked 50-m line transect oriented upslope through the plot center. Dead woody biomass and forest floor (litter and duff) depth were measured on a permanently marked 15.2-m planar transect in a random direction from each plot center [33].

We used a modification of the modified Whittaker plot [34] to describe understory vegetation. A 10 by 50-m belt transect was centered over each 50-m line transect. All species of herbaceous plants and woody shrubs were recorded in the belt. We established four 1-m² (0.5 by 2 m) subplots within each belt transect at 14-m intervals with the 2-m side parallel with the transect. Subplots were located in the top left (0–2 m) and bottom right (48–50 m) corners of the belt transect and on the right side of the line transect at 16–18 m and the left side at 32–34 m. For each subplot, we estimated the percent cover of each understory species, excluding interspaces, to the nearest 0.25% using cardboard cutouts of known sizes as visual guides. Estimates can total >100% because we estimated percent cover independently for each species and independent of canopy position.

## 2.3. Statistical Analysis

We used analysis of variance (ANOVA) for repeated measures to test for treatment effects over time on herbaceous cover (%), tree density (trees ha$^{-1}$), tree canopy cover (%), crown base height (m), shrub density (stems ha$^{-1}$) (including sprouting tree species, such as *Acer glabrum* and *P. tremuloides*), conifer regeneration (>40 cm and ≤137 cm in height; seedling ha$^{-1}$), surface fuels (sum of forest floor and 1–100-h time-lag surface fuels classes (Mg ha$^{-1}$), and canopy fuel loading (kg m$^{-2}$) (see Roccaforte et al. [35] for how CFL was estimated). Depths of the forest floor were converted to estimated forest floor loads developed from regression equations in Ffolliot et al. [36]. Prior to ANOVA, we used Shapiro–Wilk to test for data normality, and Levene's test to assess for equal variance. Regeneration density was 1/× (reciprocal) transformed to meet ANOVA assumptions for normality. When we identified time × treatment interactions, we performed one-way ANOVA tests within treatment years. For these tests, we used Tukey's HSD (honestly significant difference) post-hoc tests for pairwise comparisons of treatment group means. Changes over time from 2003 to 2018 within treatment for individual species were tested with matched paired Wilcoxon signed-rank tests. Species differences among treatments were tested with a Kruskal–Wallis test and post hoc Wilcoxon's 2-sample test for pairwise comparisons. All tests were conducted at α = 0.05. We calculated species richness as the number of species within a given plot and used the Shannon–Weiner diversity index, which uses species richness and each individual species' relative abundance, to quantify diversity [37]. Values typically range from 1.5–3.5, rarely exceeding 4.0; a higher diversity index value indicates higher plot diversity. The Shannon–Weiner diversity index considers all species equal and does not weigh sensitive, endemic, keystone, or endangered species differently. As a result, Spellerberg and Fedor [37] recommend using diversity along with another species' measurement to more fully quantify diversity. We used indicator-species analysis [38], which uses species richness and associated abundance values of species, as an additional measure to identify species that were particularly faithful (i.e., consistent indicators) for a specific forest restoration treatment. A comparison between the maximum indicator value (0–100) and random trials for occurrence of a given species (1000 Monte Carlo randomizations) provided an approximate alpha value [37]. Species with alpha levels of 0.05 or less and indicator values (INDVAL) >25 (INDVAL = relative abundance relative frequency; INDVAL ranges from 0–100) were accepted as indicator species for a specific treatment [39].

We used non-metric multidimensional scaling (NMS) to illustrate differences of the midstory plant community (shrubs and conifer regeneration (>40 cm and ≤137 cm in height)) for each experimental unit [40]. The NMS ordination was determined using 20 species (after 5% filter) with a 2-dimensional

solution. We used the Bray–Curtis distance measure, random starting configurations, 500 runs with real data, a maximum of 500 iterations per run, and a stability criterion of 0.00001. A Monte Carlo test with 9999 randomizations was used to determine how likely the observed stress value of the final solution would be by chance alone. Permutational multivariate analysis of variance (PERMANOVA) was used to quantify differences in the herbaceous understory and prostrate shrub communities among treatments and differences in the midstory (shrubs and tree regeneration) plant communities among treatments [41]. PERMANOVA uses common ecological distance measures (Bray–Curtis for this study) to examine multivariate data sets and calculates alpha values using permutations rather than tabled alpha values that assume normality. We used a one-fixed-factor design with the treatment as our main effect for the herbaceous understory and prostrate shrub community analysis and a two-factor design with treatment and time for midstory data (PC-ORD software version 5.10, McCune and Mefford [42]). We analyzed species that were present in a minimum of 5% of the plots as recommended by McCune and Grace [38].

## 3. Results

### 3.1. Pretreatment Conditions

Prior to the implementation of restoration treatments, there were no significant differences among treatments for herbaceous understory and prostrate shrub richness ($p = 0.96$), diversity ($p = 0.55$), or cover ($p = 0.35$). In addition, there were no indicator species for specific restoration treatments and PERMANOVA analysis showed no significant differences ($p = 0.64$) in the herbaceous understory and prostrate shrub communities among treatments. We also found no difference in the pretreatment means for overstory variables, shrub density, conifer seedling density, surface, and canopy fuels ($p > 0.05$) among restoration treatments. Pretreatment sampling occurred in 2003 when there was below average annual precipitation (21% deviation) and above average maximum temperature (11% deviation) (Figure 1).

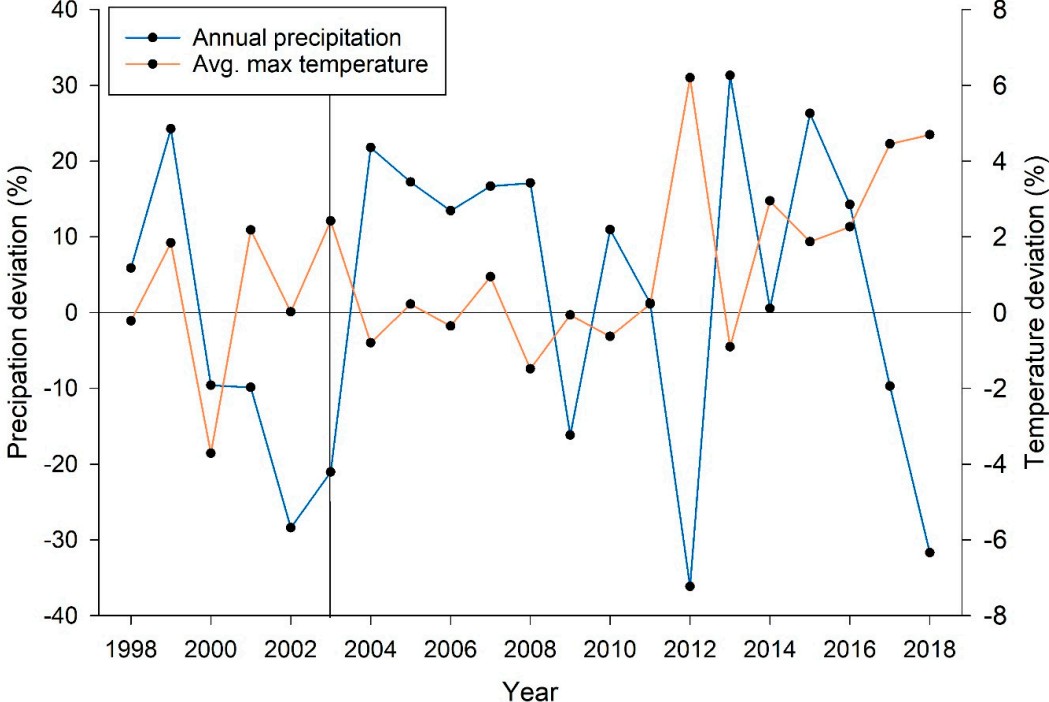

**Figure 1.** Precipitation (left axis) and temperature (right axis) percent deviation (deviation from long-term mean 1950 to 2018) spanning from 1998 to 2018 from Vallecito dam (https://wrcc.dri.edu/cgi-bin/cliMAIN.pl?co8582) near Pagosa, Colorado, USA. The vertical line represents pretreatment sampling.

### 3.2. Overstory Structure

Several forest structure variables showed a significant treatment effect as well as a significant time and treatment × time interaction (Figure 2). We found significant differences in tree density, canopy cover, and crown base height (CBH) among treatments for each post-treatment year when we analyzed measurement years separately (Figure 2). Between 1 and 10 years post-treatment, the overstory dynamics in the control illustrated slightly higher values for all variables (Figure 2). Tree density decreased after thin/burn and burn treatments, resulting in a significant difference among each treatment 1, 5, and 10 years post-treatment ($p < 0.001$). Canopy cover decreased following the thin/burn treatment, resulting in a long-term significant difference in the thin/burn compared to the control and burn treatments ($p = 0.001$). The burn treatment did not result in long-term differences in canopy cover when compared to the control.

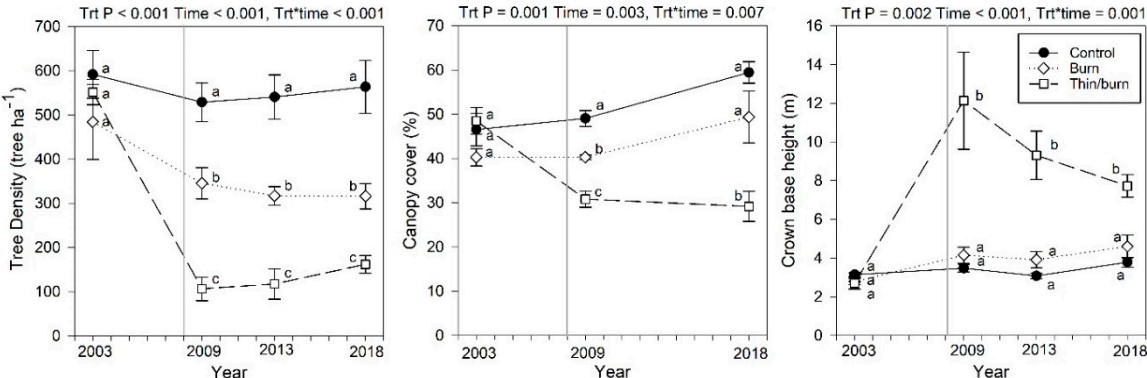

**Figure 2.** Forest structure (trees taller than 137 cm and >2.5 cm dbh) through time for both treatments and control. Bars represent 1 standard error of the mean ($n = 4$). The vertical line denotes when all treatments were completed (2008). Data comparisons are among treatments and different letters denote a significant treatment effect within a year ($p \leq 0.05$).

### 3.3. Surface and Canopy Fuels

We found no statistical differences among treatments in the mean surface fuel loading of the combined forest floor and 1–100-h time lag classes, although there was a significant time effect (Figure 3). In general, surface fuels increased over the study period (2003–2018) regardless of the treatment. By 10 years post-treatment, average surface fuels increased by 156%, 85%, and 162%, respectively, in the control, burn, and thin/burn treatments when compared to the pretreatment values.

We found no long-term differences in CBH between the burn and control treatments. CBH was significantly higher in the thin/burn compared to the burn and control treatments within each post-treatment year ($p = 0.002$). Initially, one year post full treatment, the thin/burn treatment showed an increase in CBH from 2.7 m to 12.1 m, but it decreased steadily over time, with a CBH average of 7.7 m 10 years post-treatment. Crown fuel load (CFL) showed a significant treatment effect as well as a significant time and treatment × time interaction. CFL decreased after thin/burn treatment, resulting in a significant difference compared to the control and burn treatments 1, 5, and 10 years post-treatment ($p < 0.001$). Ten years following treatment, CFL in the thin/burn treatment was 75% that of the control, and the CFL means in the burn treatments were 24% of the control with significant differences only between the control and thin/burn treatments (Figure 3).

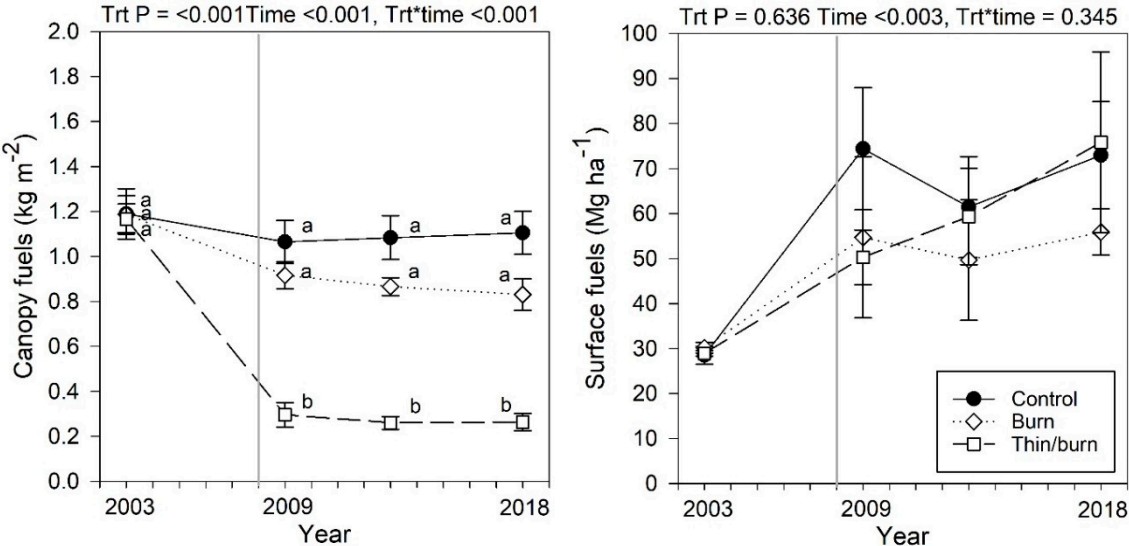

**Figure 3.** Mean canopy fuels (crown fuel load) and surface fuels (sum of forest floor and 1–100-h time lag classes) plotted through time for both treatments and control. Bars represent 1 standard error of the mean (*n* = 4). The vertical line denotes when all treatments were completed (2008). Data comparisons are among treatments and different letters denote a significant treatment effect within a year (*p* ≤ 0.05).

### 3.4. Understory Community Dynamics

Repeated measures ANOVA showed no significant main effects of treatment on the herbaceous and prostrate shrub richness (*p* = 0.91) or cover (*p* = 0.51), and no significant treatment × time interactions (0.45 and 0.51, respectively) (Figure 4). We found a significant treatment × time interaction for diversity (*p* = 0.05), and by 2018, diversity was approximately 14% higher in the thin/burn treatments compared to the control and burn, albeit not statistically significant. The 1- (2009) and 10-year (2018) post-treatment sampling periods had lower than average annual precipitation (16% and 32% deviation, respectively) and average or above average maximum temperature (0% and 25% deviation, respectively) (Figure 1). Ten years following treatment, >99% of all plant species within both treatments and the control were native in origin, with minimal, <0.1%, average non-native understory plant cover. *Cirsium arvense* was the most prevalent non-native species and was found on 10% (5%, 21%, and 4%, respectively, in the control, thin/burn, and burn plots) of the sampling plots. All understory univariate variables varied through time, though each variable responded differently over the 15-year study period. For example, species richness showed an initial increase year one post-treatment for both treatments and the control followed by a decrease 10 years post-treatment (Figure 4). Diversity for all treatments slightly increased 1 year post-treatment followed by a continued increase 10 years post-treatment for the thin/burn and a decrease in the control and burn only treatments (*p* = 0.01; Figure 4). Finally, herbaceous/prostrate shrub cover increased by 3%, 20%, and 79%, one year following treatments, followed by an increase of 35%, 33%, and 5% in 2018 (respectively in the control, burn, and thin/burn treatments). Over the 15-year study period, increases in the herbaceous/prostrate shrub cover were driven by the doubling of graminoid cover across all treatments and a 64% increase in the perennial forb cover in the thin/burn treatment. By 2018, graminoid cover averaged 10.7% in the control and 9.7% in both the burn and thin/burn treatments. Perennial forbs averaged 8.9%, 8.8%, and 11.5%, respectively, in the control, burn, and thin/burn treatments.

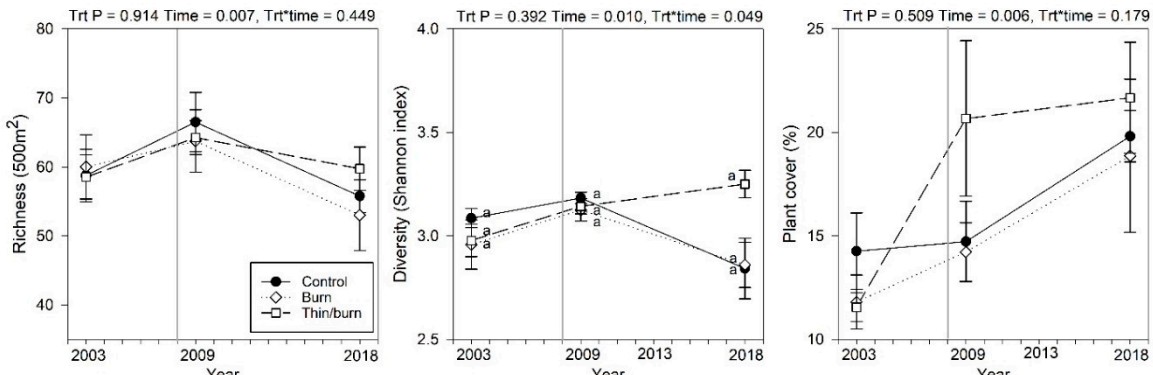

**Figure 4.** Mean herbaceous/prostrate shrub richness, diversity, and cover plotted through time for both treatments and the control. Bars represent 1 standard error of the mean (*n* = 4). The vertical line denotes when all treatments were completed (2008). Data comparisons are among treatments and different letters denote a significant treatment effect within a year (*p* ≤ 0.05).

When we analyzed the graminoid community data separately from the herbaceous understory and prostrate shrub community data, there were no differences in the graminoid community among treatments as indicated by PERMANOVA (*p* = 0.55), but there were significant differences across time (*p* = 0.00001) for all time comparisons (pre to post-one1 pre to post-10, and post-1 to post-10) regardless of the treatment. There were no graminoid indicator species for pretreatment or one year post-treatment, but there were five C3 graminoid indicator species for 10 years post-treatment: *Poa pratensis*, *Elymus elymoides*, *Carex geophila*, *Poa fendleriana*, and *Bromus* spp. (Table 1).

**Table 1.** Herbaceous/prostrate shrub indicator species associated with restoration treatments in 2018, 10 years post-treatment, and graminoid indicator species associated with time. We accepted species with indicator values (relative abundance x relative frequency) >45 and *p* values ≤0.05 as indicator species for a specific treatment (control, burn, thin/burn) or time (2003-pretreatment, 2009-1 year post-treatment, and 2018-10 years post-treatment). There were no 2003 or 2009 indicator graminoid species. *Bromus* spp.* includes *B. ciliatus*, *B. lanatipes*, and <5% *B. inermis*.

| Treatment/Time | Species | Indicator Value | *p*-Value |
|---|---|---|---|
| **TREATMENT** | | | |
| Control | *Pedicularis racemosa* | 75 | 0.05 |
| Control | *Erigeron divergens* | 74.6 | 0.017 |
| Burn | *Paxistima myrsinites* | 74.5 | 0.035 |
| Thin/Burn | *Chamerion angustifolium* | 83.3 | 0.02 |
| Thin/Burn | *Cirsium arvense* | 80.6 | 0.02 |
| Thin/Burn | *Vicia americana* | 69.5 | 0.007 |
| Thin/Burn | *Galium boreale* | 58.4 | 0.007 |
| Thin/Burn | *Gentianella amarella* | 46.8 | 0.017 |
| **TIME** | | | |
| 2018 | *Poa pratensis* | 83.1 | 0.0002 |
| 2018 | *Elymus elymoides* | 78.6 | 0.0008 |
| 2018 | *Poa fendleriana* | 75.1 | 0.0004 |
| 2018 | *Carex geophila* | 68.0 | 0.0002 |
| 2018 | *Bromus* spp.* | 56.4 | 0.005 |

The herbaceous understory and prostrate shrub communities 10 years post-treatment were significantly different among treatments using PERMANOVA analysis. Specifically, there were differences between the control and thin/burn treatments (*p* = 0.03) and burn and thin/burn treatments (*p* = 0.03). There were two native perennial forb indicator species, *P. racemosa* and *E. divergens*, in the control and one native prostrate shrub indicator species, *P. myrsinites*, for the burn treatment

(Table 1). There were five forb indicator species in the thin/burn treatment: Three native perennial forbs, *C. angustifolium*, *V. americana*, and *G. boreale*; a non-native perennial forb, *C. arvense*; and one native biennial, *G. amarella* (Table 1).

### 3.5. Midstory Shrubs and Conifer Regeneration

The shrub midstory consists of vegetation 50 cm–8 m in height (Short 1986). Shrubs and conifer seedling densities responded strongly to inter-annual climatic differences, but there were significant treatment effects for shrub and conifer seedling densities, as well as a significant treatment × time interaction for conifer seedlings (Figure 5). By 2018, 10 years post-treatment, shrub densities in thin/burn were on average 24% and 34% greater than the control and burn, respectively. Over the 15-year study period (2003–2018), mean shrub densities increased by 86%, 76%, and 144%, respectively, in the control, burn, and thin/burn treatments (Figure 5). Furthermore, the size classes' distribution shifted toward taller shrubs, being especially pronounced in the thin/burn treatment (Figure 6). For example, in 2003, prior to treatments, the proportion of shrubs >80 cm in height was <14% across all treatments and by 2018 this proportion increased to 30%, 27%, and 41%, respectively, in the control, burn, and thin/burn treatments. *A. alnifolia*, *Q. gambelii*, and *S. rotundifolius* were the dominant species driving the total shrub density increases between 2003 and 2018 (Table 2).

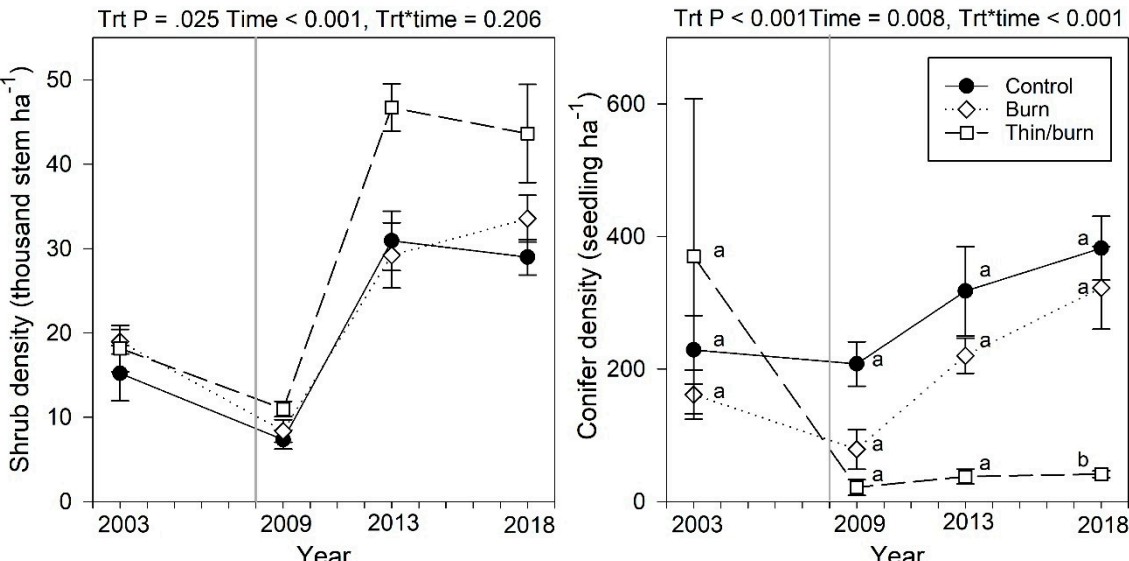

**Figure 5.** Mean shrub and conifer seedling density (>40 cm and ≤ 137 cm in height) plotted through time for both treatments and control. Bars represent 1 standard error of the mean (*n* = 4). The vertical line denotes when all treatments were completed (2008). Data comparisons are among treatments and different letters denote a significant treatment effect within a year (*p* ≤ 0.05).

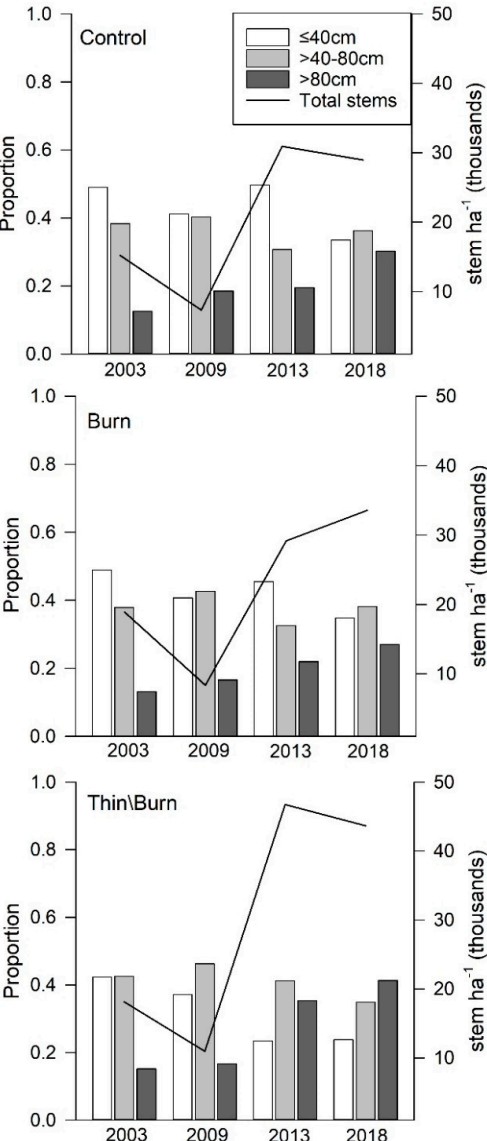

**Figure 6.** Proportion (bars and left *y*-axis) of the total stem density (line and right *y*-axis) of shrubs within three height classes in 2003 (pretreatment) and three post-treatment measurement years (2009, 2013, 2018). Each panel represent a treatment type (*n* = 4).

Conifer seedling densities showed a main effect of treatment as well as a significant time × treatment interaction (Figure 5). When analyzed separately within years, we found that conifer seedling densities did not differ among the three treatments 1 (*p* = 0.21) or 5 years post-treatment (*p* = 0.06); however, by 10 years post-treatment, conifer seedlings were significantly lower (*p* < 0.01) in the thin/burn treatments compared to the control and burn. Ten years post-treatment, total conifer seedling density differences among treatments were driven by *A. concolor*, *P. ponderosa*, and *P. menziesii* (Table 2).

Midstory community composition indicated a significant treatment effect as well as a significant time effect but no significant treatment × time interaction (Figure 7). There were four indicator species reflecting a dominance in the thin/burn treatment in 2013: *P. virginiana* (IV = 71.1, *p* = 0.04), *P. tremuloides* (IV = 62.2, *p* = 0.04), *Q. gambelii* (IV = 52.5, *p* = 0.03), and *R. woodsii* (IV = 49.4, *p* = 0.01). There were also two indicator species in the control treatment in 2013: *P. menziesii* (IV = 72.8, *p* = 0.01) and *A. concolor* (IV = 61.7, *p* = 0.04).

**Table 2.** Average density and frequency of the most common midstory species in 2003 (pretreatment) and 2018 (10 post-treatment) by treatment. N = 4; $p \leq 0.05$. Treatment differences for density (within year) are indicated by letters (a, b for 2003 and x, y, z for 2018). Pretreatment (2003) and 10 post-treatment (2018) species density differences within treatments are indicated by bold text.

| Species | Year | Density (Stems/Trees ha$^{-1}$) | | | Frequency (Proportion of Plots) | | |
| --- | --- | --- | --- | --- | --- | --- | --- |
| | | Control | Burn | Thin/Burn | Control | Burn | Thin/Burn |
| Density (stems ha$^{-1}$) | | | | | | | |
| AMAL | 2003 | **3516.3** | 4317.5 | 4712.5 | 0.93 | 0.91 | 0.93 |
| | 2018 | **6565.0** | 8190.0 | 10,205.0 | 0.84 | 0.89 | 0.86 |
| PRVI | 2003 | 251.3 | 433.8 | 885.0 | 0.20 | 0.28 | 0.50 |
| | 2018 | 398.8 | 496.3 | 1501.3 | 0.15 | 0.28 | 0.55 |
| QUGA | 2003 | **3666.3** | 4598.8 | **4426.3** | 0.75 | 0.71 | 0.85 |
| | 2018 | **6222.5** | 7670.0 | **10,080.0** | 0.68 | 0.70 | 0.83 |
| ROWO | 2003 | 1952.5 | 1613.8 | **1563.8** | 0.74 | 0.63 | 0.80 |
| | 2018 | 3495.0 $^{x,y}$ | 2325.0 $^y$ | 4725.0 $^x$ | 0.65 | 0.59 | 0.78 |
| SYRO | 2003 | **5785.0** | 7638.8 | **6413.8** | 0.99 | 1.00 | 0.95 |
| | 2018 | **12,170.0** | 11,630 | **15,805** | 0.99 | 0.93 | 0.99 |
| ABCO | 2003 | 166.3 | 107.5 | 347.5 | 0.51 | 0.43 | 0.60 |
| | 2018 | 108.8 $^x$ | 63.8 $^x$ | 11.3 $^y$ | 0.36 | 0.26 | 0.06 |
| PIPO | 2003 | 3.8 | 8.8 | **6.3** | 0.04 | 0.05 | 0.04 |
| | 2018 | 163.8 $^x$ | 223.8 $^x$ | 25.0 $^y$ | 0.40 | 0.39 | 0.11 |
| POTR | 2003 | 856.3 | 493.8 | 308.8 | 0.63 | 0.35 | 0.39 |
| | 2018 | 837.5 | 682.5 | 1411.3 | 0.68 | 0.40 | 0.43 |
| PSME | 2003 | **58.8** | 45.0 | 16.3 | 0.28 | 0.21 | 0.13 |
| | 2018 | **108.8** $^x$ | 31.3 $^y$ | 5.0 $^z$ | 0.43 | 0.18 | 0.03 |

AMAL = Amelanchier alnifolia, PRVI = Prunus virginiana, QUGA = Quercus gambelii, ROWO = Rosa woodsii, SYRO = Symphoricarpos rotundifolius, ABCO = Abies concolor, PIPO = Pinus ponderosa, POTR = Populus tremuloides, PSME = Pseudotsuga menziesii.

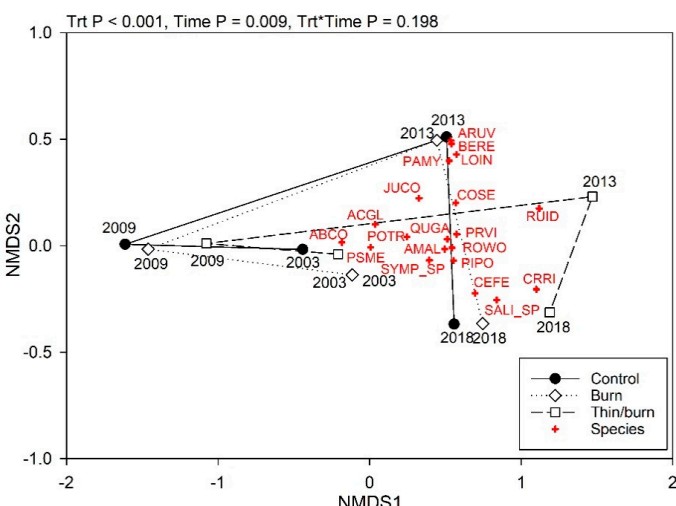

**Figure 7.** Non-metric multidimensional scaling (NMS) ordination of the midstory plant community (shrubs and conifer regeneration (>40 cm and ≤ 137 cm in height)). Each symbol represents a treatment within a sampling year. The final solution had 2 dimensions and represented >90% of the variation of the Bray–Curtis distance matrix (stress 9.4, $p = 0.0001$). PERMANOVA results are presented above the graph ($p \leq 0.05$). ACGL = *Acer glabrum*, AMAL = *Amelanchier alnifolia*, ARUV = *Arctostaphylos uva-ursi*, BERE = *Berberis repens*, CEFE = *Ceanothus fendleri*, COSE = *Cornus sericea*, JUCO = *Juniperus communis*, LOIN = *Lonicera involucrata*, PAMY = *Paxistima myrsinites*, PRVI = *Prunus virginiana*, QUGA = *Quercus gambelii*, ROWO = *Rosa woodsii*, RUID = *Rubus idaeus*, SALI_SP = *Salix* sp, SYRO = *Symphoricarpos rotundifolius* ABCO = *Abies concolor*, PIPO = *Pinus ponderosa*, POTR = *Populus tremuloides*, PSME = *Pseudotsuga menziesii*.

## 4. Discussion

### 4.1. Overstory Dynamics

Numerous studies have illustrated significant shifts in forest structure resulting from forest restoration treatments in dry forest types [9,23,43–47]. Researchers have found that the most significant effects on forest structure (i.e., stand density, tree canopy cover, and CBH) come from treatments that include some type of harvest combined with prescribed fire [23,44,48–51]. Restoration and fuel reduction treatments are designed to change numerous forest stands and ecosystem properties, such as forest species composition, diameter distributions, and tree vigor [49]. In addition, researchers have shown that thin/burn treatments have the highest resistance to simulated active and passive crown fire [44,50,51].

Similar to other studies, we found that thin/burn treatments showed the greatest reduction in stand density, tree canopy cover, and CBH, but these trends started to diminish 10 years post-treatment. For example, stand density 10 years post-treatment was 162 trees/ha, an increase from 106 trees/ha one year post-treatment but still within the 1870 reconstructed reference conditions of 141 trees/ha for our study site [6]. Similar to stand density, canopy cover (29%) in the thin/burn 10 years post-treatment was within the midrange of the historical mean canopy cover reference conditions (21%–43%) established for other mixed-conifer forests in the region [52,53]. However, the burn and untreated controls showed structural characteristics similar to one another, indicating the low effectiveness of the burn treatment for altering stand characteristics and meeting restoration goals. This finding contrasts with a study by Stephens et al. [45], which showed that prescribed fire did significantly alter forest stand structure and fuel dynamics in mixed-conifer forests of the Sierra Nevada range in California seven years post-treatment. This disparity likely reflects variations in prescribed fire behavior; therefore, land managers need to align burn conditions with treatment objectives when planning restoration. While the density in the thin/burn treatment was still significantly lower than pre-treatment (677 trees/ha—Fulé et al. [6]), density will likely continue to increase over time without additional maintenance burning as shade-tolerant white fir and Douglas fir saplings attain overstory stature. Some researchers have advocated for higher stand densities to buffer against potential mortality that may occur following harvest or prescribed fire [54,55]. However, the thin/burn stand density in our study stayed neutral or increased over time following treatment.

### 4.2. Treatment Effects on Understory Vegetation

Understory vegetation responses to forest restoration treatments in dry forest types are complex and vary at spatial and temporal scales due to parameters, such as treatment intensity, climate, and biotic (e.g., overstory and understory vegetation, seed availability, soil microbes, mycorrhizae, competitive interactions), and abiotic conditions (e.g., light, nutrients, surface fuels, soil substrate availability) [15,20,22,56–60]. In contrast to other research studies that illustrated differences among treatments for species richness, diversity, and cover in dry forest types (e.g., [15,17,58,60,61]), we did not find any significant treatment effects in these variables. This disparity may be the result of differences in study methodologies, spatial and temporal scales, or site conditions [15,59]. In our study, we found a significant treatment × time interaction for diversity and significant time effects for species richness, diversity, and plant cover. Time effects and interannual variation in understory responses have been reported for southwestern forest types, where climate can strongly influence the distribution and performance of plants [60,62–64].

Although we found no treatment effects on understory cover, graminoid cover doubled across both treatments and controls over the 10-year study period and led to overall increases in herbaceous cover. Springer et al. [22] also noted that a doubling of graminoid cover for treated and untreated sites in mixed-conifer following wildfire in northern Arizona and New Mexico contributed to significant plant cover increases over a 4-year period. The authors attributed increases in graminoids to the ability of wind-dispersed species to colonize disturbed areas rapidly and tolerate xeric conditions under an

open forest canopy [22]. In our study, C3 graminoids, specifically five species, were responsible for the majority of the graminoid cover increase across both treatments and control. In both treatments and control, *P. pratensis* had the highest graminoid indicator value 10 years post-treatment. Similarly, Strahan et al. [60] illustrated a significant three-fold increase of *P. pratensis* cover across both treatments and controls in another restoration study in the southwest over a 12-year period and Springer et al. [22] highlighted that *P. pratensis* had the third highest graminoid indicator value 5 years post-wildfire in their study. The other four C3 graminoid indicator species in our study were also present in the study by Strahan et al. [58] but did not show similar patterns of cover increases across time. However, in a study by Kerns and Day [59], the authors saw an increase across the entire study site for *E. elymoides*, which is rhizomatous and a rapid colonizer following disturbance (Moore et al. 2006). Likewise, Springer et al. [22] also saw significant increases in two of our other four graminoid species, *C. geophila* and *E. elymoides*, over a 5-year period regardless of the treatment.

Our findings suggest that the understory dynamics in these forests are complex and other variables, such as climate and competition, may play roles on par with thinning and prescribed fire in determining understory responses. In our study, precipitation was below the long-term average, and the maximum annual temperature was similar to or above its long-term average, in all three understory sampling years. These conditions may have contributed to lower understory cover gains in the more open thin/burn treatment. More research is needed to tease apart understory vegetation responses and climate effects to better understand forest restoration treatments and inform managers' expectations given a warmer drier climate [65].

One common understory vegetation response to forest restoration treatments is an increase in non-native plants associated with thinning and/or prescribed fire treatments [20,21,57,66]. In our study, prior to restoration treatments, there was extremely low (<1%) non-native understory species richness and cover (<0.1%), likely due to the minimal historical disturbances at our study area, which included one limited tree harvest and restricted grazing [8]. This finding is consistent with other studies in ponderosa pine forests, where non-native plants did not increase following restoration treatments [22,67]. A lack of a significant treatment effect on understory cover in our study may be the result of high pre-existing spatial variability within individual restoration treatment units prior to treatment that remained following treatments. In addition, strong shrub responses may have inhibited herbaceous establishment and growth.

### 4.3. Future Resiliency

One of the main objectives for fuel reduction treatments is to decrease connectivity between surface and canopy fuel layers and reduce the chance for initiation and propagation of crown fire [49,68,69]. Of all the forest structure variables in our study, CBH had the largest change between 1 and and 10 years post-treatment in the thin/burn. Specifically, CBH went from over a 300% increase in the thin/burn one year after treatment to almost a 100% decrease 1 to 10 years post-treatment. During the same time period, CBH slightly increased in the control and burn 1 to 10 years post-treatment. The significant increase in CBH in the thin/burn 1 year post-treatment is consistent with numerous other studies [44,45,49,50]; however, the decrease in CBH over the following 9 years is concerning from a fuel reduction standpoint.

Restoration treatments that include some type of harvest generally show a reduction in CFL [44–46,48,50,70]. Indeed, we saw a 75% significant decrease in CFL in the thin/burn treatment compared to the control 10 years post-treatment, with no significant difference in CFL for the burn treatments compared to controls. Tree harvest also may have an immediate increase in surface fuels [23,28,44,45] while subsequent burning may then reduce surface fuels by as much as 90% [49,71]. Without prior thinning, first-entry prescribed fire may increase surface fuels due to mortality and the fall of small trees [45]. In our study, surface fuels increased immediately across both treatments and controls, then continued to increase across the monitoring period, with no significant differences between treatments. These additional fuels most likely are the result of small trees dying due to

competition or disease in our controls, small harvested trees lopped and scattered in our thin/burn treatments, and small fire-killed trees in burn treatments. Another possible input of surface fuels is shrub biomass. Midstory vegetation, such as tall shrubs, plays an important role in crown fire initiation and support because it serves as a transition fuel (i.e., ladder) between the surface and canopy fuels. Thus, one of the most effective ways to alter fire behavior is to break the continuity of the surface, ladder, and crown fuels [69,72]. We found a >250% increase in shrub density over our 10-year monitoring period in treatments as well as the control, suggesting climate was an important driver of shrub responses (Figure 8). In addition, over the 10 years post-treatment, we saw an increase in the average shrub height for both treatments and the control, with shrub stems >80 cm becoming the dominant size class in the thin/burn treatment. The overall increase in both the shrub density and mean height class for both treatments and controls over 15 years illustrates that our stands, regardless of the treatment, became more susceptible over time to crown fire behavior due to the transitional fuel source between the surface and forest canopy.

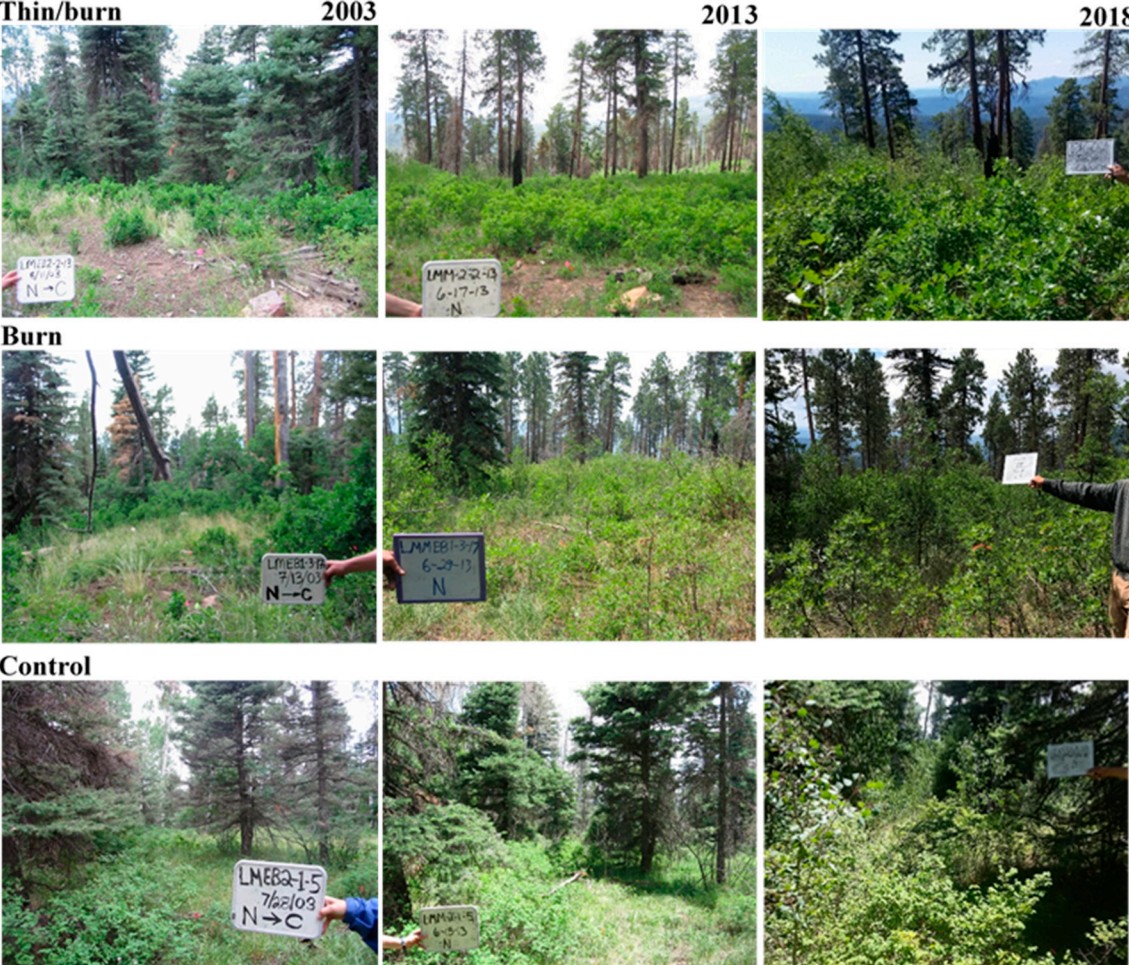

**Figure 8.** Repeat photos taken from alternative restoration treatments before treatment (2003), 5 years after treatment (2013), and 10 years after (2018).

The three most dominant shrub species at our site were *S. rotundifolius*, *Q. gambelii*, and *A. alnifolia*, all of which resprout after fire [73,74]. In addition, *Q. gambelii* has a high drought tolerance due to deep roots, xeromorphic leaves, and efficient water transport [75–77]. The immediate shrub response to thin/burn and burn treatments is commonly a decrease in stem density and/or cover [18,78,79]. Our findings followed this pattern as all treatments, including the control, showed an immediate decrease in shrub density one-year post-treatment. Following this decrease, however, we saw a

significant increase 5 years post-treatment for both treatments and controls, with significantly higher shrub density in the thin/burn treatments than burn or controls, which is similar to other findings [11]. Previous research has illustrated that shrub density begins to decrease as shrub height increases [80,81]. Approximately 40% of all shrub stems were >80 cm 10 years post-treatment in our thin/burn treatments, supporting the inverse relationship between shrub density and height found in other studies.

In addition to increasing ladder fuels and inhibiting herbaceous plant responses, shrub density in our study may also have limited conifer regeneration in the thin/burn treatment. Numerous studies have quantified short-term effects of shrubs on conifer regeneration [17,18], but fewer studies have investigated long-term trends [11,78]. Conifer seedlings described in this study generally decrease immediately following thinning and prescribed fire treatments either from direct fire mortality, a reduced number of seed-producing trees, or a lack of a fire-prepared seedbed [17,78,82]. We saw a similar response in our study, where seedling densities decreased by 51% and 94% for burn and thin/burn, respectively, 1 year post-treatment. Furthermore, 10 years post-treatment, conifer seedling densities remained significantly lower in the thin/burn compared with the control and burn treatments. Recent research has illustrated that under a warmer drier climate, forests are susceptible to shrub-type conversion without adequate conifer regeneration, especially in areas that have experienced high-severity wildfire [83–85].

## 5. Conclusions and Management Implications

A common goal for forest restoration treatments in dry forest types is to restore ecosystem function and promote resiliency to future disturbances by altering forest structure and reducing fuel accumulations through mechanical treatments and prescribed fire [6,9,12,72,86]. In our study, 10 years post-treatment for the burn only and thin/burn treatments, some of the initial treatment benefits to promote resiliency to future disturbances had already or were starting to diminish (e.g., increasing shrub density and decreasing crown base heights). In addition, our main finding of no strong long-term treatment effect on the native herbaceous understory and prostrate shrubs is consistent with several published studies [57,59]. Cover increases in the control and burn treatments between 2009 and 2018 negated the initial gains of cover in the thin/burn treatment, which in turn showed shallower gains in cover over this time period. This pattern, taken together with the vigorous midstory shrub responses, suggests that herbaceous abundance may have been inhibited by increases in the midstory shrub layer. Such responses may present a challenge to managers aiming to reduce fuel loading and potential for crown fire in these systems. More frequent monitoring of the herbaceous layer would assist in gaining a better understanding of the interactions of climate variability and competition with other vegetation layers [15,61,87]. Given the transient nature of some of our forest restoration treatment effects 10 years post-treatment, re-entry with prescribed fire is one tool that managers could use if the main management goal is to raise CBH and decrease shrub ladder fuels that can transition surface fire to the tree canopy, tree canopy cover, CFL, and surface fuels. This recommendation is consistent with Reynolds et al.'s [4] recommendation that forest restoration treatments in southwestern forests with frequent fire regimes may need multiple treatments to meet broad forest restoration goals. However, given the lack of conifer regeneration in the thin/burn treatments and the dominance of shrubs in the midstory, additional prescribed fire treatments may be a tipping point that facilitates a type conversion in the midstory from conifer to shrubs in the event of a major disturbance, such as severe drought, insect outbreak, or wildfire. Under a warmer drier climate with more high severity fire, researchers have predicted that type conversions from forest to shrub communities will become more common due to a lack of conifer regeneration [83,84]. As a result, managers will need to balance long-term restoration and fuel mitigation goals, which may conflict given the variable treatment responses we identified in our study. Overall, there is a need for additional long-term studies that quantify long-term forest restoration effects to provide land managers the information they need to allocate limited funds and resources to maintaining existing treatments or implementing novel restoration treatments to address future uncertainties [45].

**Author Contributions:** J.E.K. contributed to the organization and conceptual ideas, analyzed understory data, interpreted results and writing majority of the manuscript. M.T.S. contributed to conceptual idea, analyzed midstory and overstory data, created graphs and writing. D.W.H. contributed to editing, interpreted results and writing. All authors have read and agreed to the published version of the manuscript.

**Funding:** This research was funded by a US Forest Service grant, award #18-DG-11031600-057.

**Acknowledgments:** We would like to thank the staff and students at the Ecological Restoration Institute at Northern Arizona University and Fort Lewis College and the San Juan National Forest for field and logistical support. We thank Michael Battaglia and Matt Tuten for insightful comments on an earlier manuscript draft.

**Conflicts of Interest:** The authors declare no conflicts of interest.

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
