# Peer review of "Effectiveness of Restoration Treatments for Reducing Fuels and Increasing Understory Diversity in Shrubby Mixed-Conifer Forests of the Southern Rocky Mountains, USA"

_forests, doi:10.3390/f11050508_

Round 1

Reviewer 1 Report

Dear authors,

I really enjoyed reading your paper. There is a worldwide need of long-term studies quantifying long-term forest restoration effects, especially under the risk of climate change and alterations in fire regimes. The manuscript is well written and logically structured. I have only some minor comments and suggestions (see attached pdf). Nevertheless, I strongly suggest you to add a map of the study site as figure 1.

Reviewer 2 Report

General comments

  The authors have tackled important questions related to the ecological restoration of  mixed-conifer  forests dominated by ponderosa pine. Although the scientific literature on this topic is abundant there is still a substantial lack of information on the long-term effects of the restoration treatments, extensively implemented by forest managers in the last decades in the US southwestern conifer forests. This study is a new contribution to partially fill this knowledge gap by providing data for a ten years period after treatments application and thus helping us to gain perspective on if the restoration treatments have met their planned objectives. In that sense long-term studies like this, with data collected in field experiments, are certainly welcome.

 The authors reveal a good knowledge of the ecosystem under study and of the literature and results obtained in previous research on the subject.

The study was well designed, and it has been developed in a well known experimental area where previous quality research on this topic has been conducted. Methods used are appropriate and the information generated has been treated statistically on a solid basis. As in other similar field studies using combined treatments, the lack of completing the planned treatments in the same year for all blocks, practically does not imply a serious handicap, especially in a ten years perspective. Also, it seems more reasonable than to statistically treat the information separately for the blocks burned in 2007 and 2008 because that would have significantly reduced the power of the ANOVA in an experiment with great variability between blocks. This is a common problem in similar forest field experiments.

This study has paid more attention to the response of ground vegetation than to fuels, perhaps to avoid lengthening the text or because the authors prefer to address this issue in a separate study. For instance, understory biomass is not evaluated although the reader can obtain a rough idea of the treatments effect on the structure of this very influential fuel layer for the risk of canopy fires. Presentation of tree height measurements over time, combined with crown fuel loads, would have allowed the reader to make a clearer picture of the variation of other fuel attributes such as crown bulk density(CBD)  and thus to ponder the shift in active crown fire hazard caused by the treatments. Therefore it is not easy to estimate whether thin/burn treated stands  are less prone to active crown fires although the detected understory encroachment, decreased canopy cover and tree density suggest lower dead fuel moisture and higher under canopy wind  velocity potentially leading to higher surface fire linear intensity.

The authors have well expressed some of the dilemmas that managers of restoration treatments face and this gives an added quality to the work done. At first glance, and taking all results together, it seems that there has been some difficulty in achieving all objectives foreseen in that restoration treatment. For instance, the authors underscore the marked and worrying trend to decrease canopy base height (CBH) in thin/ burn treatment over time along with a higher understory biovolume increase (although with no change in shrub cover) and a dramatic decline in conifer seedling density.  However, the fact that ten years after thinning, CBH is still twice as high as the other treatments does not seem to be an unfavorable result from the crown fire hazard reduction perspective. In fact, it would seem quite predictable that some time after the heavy thinning trees has taken advantage of the increased availability of light, to emit new low branches. Likewise that understory has grown more vigorously as competition with the tree is reduced and more light, water and nutrients has been available, and in turn competes more with conifer seedlings.

Regarding those aspects the authors could reconsider more explicitly whether current restorative treatments can be improved. For instance, could a first lighter thinning be more advantageous, taking into account that the stands already had a moderate density initially. Apparently the long fire exclusion period, the altered tree specific composition and the initial diameter distribution have made the restoration too difficult to be carried out in a single intense (80% of tree density reduction)  intervention (thin/burn treatment). Climate change projections point to increased dryness in the area. On the one hand this seems to suggest that a greater intensity of thinning of shade tolerant species would be desirable to limit water stress, but on the other hand, could not induce that a change in fuel and microclimatic conditions jeopardizing restoration objectives in terms of regeneration, return to the historical fire regime and fire resilience increase? In short, couldn't a more gradual thinning with more frequent and therefore less intense prescribed fires be more appropriate and allow the survival of conifer seedlings?

Overall, in my opinion, the study provides insight into a complex problem, and discover new gaps in both knowledge and sustainable management. The authors have made a valuable contribution, based in new detailed field information and quality data.

 Some minor changes are suggested below. In my opinion after the completion, the manuscript should be accepted.

Special comments

Abstract

    Please, include a paragraph stating restoration treatments did not affect non -native plants

Line 24. Please, include a sentence indicating that the shrub distribution changed toward taller height classes in thin/burn

  1. Materials and methods. Botanical name of species need to be written in cursive.
  2. Results.

Lines 214, 216 and 235 .Please, change ”immediately” to “ on year post full treatment”

 Lines 225 and 226 and Fig.3 . The authors do not clarify how forest floor fuel load is determined. In the materials and methods section they indicated that they used Brown's (1974) methodology to determine the load of downed and dead woody  fuels and the thickness of the forest floor (lines 140-142). That method allows calculating the load of the woody debris by using field inventory data and standard values of specific gravity, but does not report values of bulk density of forest floor. I understand that the authors have used some value for this last property and thus to convert the values of forest floor depth in loads, but to indicate the source is necessary.  Please, comment this point.

  1. Discussion

Lines 413-420

Some comment on the possible effect of the 3-4 years delay between thinning+slash scattered on the ground  and burning on the understory vegetation response and if that is the usual case would be welcome.  

Please, check the numbering of the references that do not correspond to those in the manuscript. The reason seems to be that the  reference 22 ( line 591 ) was eliminated
